# A Study on the Impact of Small-Scale Courtyard Landscape Layouts on Spatial Oppressiveness in Dense High-Rise Environments

Ying Cao * and Lianghao Huang

School of Mechanics and Civil Engineering, China University of Mining and Technology, Beijing 100083, China; huanglianghao21@gmail.com
* Correspondence: caoying1316@163.com

**Abstract:** Numerous studies have shown that the oppressiveness brought on by high-rise buildings can be somewhat mitigated by landscapes. However, there is a lack of research that specifically examines the relationship between courtyard landscape layouts and spatial oppressiveness. This study focuses on the relationship between the landscape layout of a small courtyard and spatial oppressiveness. It entails tests that are conducted in two phases of experiments that examine visual, behavioral, and psychological aspects. In the first experiment, participants were asked to freely explore four sample scenarios without any predetermined outcome, and their behavioral coordinates were recorded as behavioral data. Using the semantic differential (SD) method, participants in the second experiment used four example panoramic landscapes to assess oppressiveness and supply psychological indicators (including oppressiveness, attractiveness, territoriality, and desire to stay). Additionally, this study quantified the visual elements' solid angles in the scenes through panoramic image segmentation. The results ultimately show that landscape layouts, particularly the surrounding and dispersed layouts, are more effective in alleviating the oppressiveness induced by surrounding buildings compared to the centralized layout. Furthermore, the study explains the process of how landscape layouts mitigate oppressiveness through visual elements and behavioral intention.

**Keywords:** oppressiveness; landscape layout; high-rise environment; behavioral analysis; greenery

## 1. Introduction

Oppressiveness is often used to describe that buildings in high-rise environments will bring negative psychological pressure to residents [1]. With the rapid development of the city, the contradiction between population growth and limited land resources becomes more and more obvious, and the high-rise environment is an inevitable trend of urban development. Courtyards, as a typical landscape space in high-rise environments, have a healing effect on residents, which has been proven by many studies [2–5]. However, the oppressive impact imposed by the surrounding high-rise buildings on the courtyard space cannot be overlooked. To safeguard the courtyard's ability to foster the physical and mental well-being of urban residents, it is imperative to conduct a comprehensive study on the factors influencing this sense of oppressiveness within the courtyard space and explore effective strategies to mitigate it.

The research on oppressiveness was mainly led by Japan, and the first theory of outdoor space openness published by Ogiso in 1971 involved the oppressiveness of metropolitan landscapes [6]. Studies on oppression typically fall into one of two categories: techniques for measuring oppressiveness or relieving oppressiveness. In the research of measurement and formula calculation for oppressiveness, Takei and Ohara [7–10] first constructed the relationship between oppressiveness and building a solid angle through research, that is, $log\psi = 0.747logS + 1.710$, where $\psi$ is oppressiveness and $S$ is building solid angle. Building upon Takei and Ohara's research, Hwang et al. [11] demonstrated

that the three-dimensional angularity of architecture can better explain the formation of oppressiveness. They established a relationship between oppressiveness, architectural three-dimensional angularity, and distance as follows: $\psi = \sum \Omega \Gamma^3$. Here, $\psi$ represents the oppressiveness, $\Omega$ stands for the architectural three-dimensional angularity, and $\Gamma$ represents the horizontal distance between individuals and the architectural interface. Furthermore, by including the shady impacts of trees on buildings and the sky in the research, Morteza et al.'s study [12] broadened the investigation of oppressiveness. They expanded the theoretical equation for measuring oppressiveness to: $\psi = \sum (\Omega_B - \Omega_{TCB})\Gamma^3$. In this equation, $\Omega_B$ denotes the three-dimensional angularity of the architecture and $\Omega_{TCB}$ represents the three-dimensional angularity of trees shading the buildings.

In the research on methods to alleviate oppressiveness, the role of landscapes has been substantiated by numerous studies. Takei's research [13] demonstrated that due to the presence of plants around high-rise buildings, oppressiveness does not increase infinitely with the height of the building. Morteza et al. [12] employed the Semantic Differential (SD) Method to experimentally study the alleviating effects of oppressiveness through two types of greenery: vertical greening of high-rise facades and the planting of trees around high-rise buildings. Their findings revealed that vertical greening had a limited impact on alleviating oppressiveness, while trees planted in front of buildings had a significant effect. In a study conducted by Xu et al. [14] in three landscape units in Hangzhou, China, the relationship between the oppressiveness, building height, and the condition of street trees was analyzed. It was concluded that street trees had a significant correlation with alleviating oppressiveness, and this alleviating effect became more pronounced with increasing building height. Cui et al. [15], in their experimental research on the external environment of buildings and their oppressiveness, reached the conclusion that high-rise buildings with street trees or facade greenery had permissible values for architectural three-dimensional angularity that were lower than the benchmark value. Specifically, these values were 65.7% and 69.6%, respectively, compared to the baseline value of 72.1%. These research findings support Ulrich's psycho-evolutionary theory [2,16], which posits that engaging with nature helps alleviate stress.

In the past, environmental factors, including plant coverings, plant species, and plant height, were largely considered in both oppressiveness study measurements as well as methods of alleviating oppressiveness. Likewise, the primary goal of the research methodology was to evaluate the connection between environmental changes and psychological reactions from a single perspective. However, the process of experiencing oppressiveness, being a perceptual outcome, is inevitably intertwined with behavior. Several phenomenological theories provide relevant insights into this matter. Heidegger contends that cognition cannot be solely understood as a relationship between a subject and an independent object. Instead, the cognitive activities of the subject should be seen as open responses to situations or the world [17]. Husserl argues that there is an inherent functional interdependence between visual perception and the bodily experience of movement, which determines the constitution of visual perception [18]. Moreover, from a behavioral perspective, Altman and Stokols suggested that humans and the environment are not isolated entities. While the environment influences human behavior, human cognition also assigns value to and reinterprets the environment [19]. These theories indicate that the process by which oppressiveness, as a psychological, cognitive reaction, is influenced by the environment is a mutual effect rather than a passive one-way transmission. Therefore, it is crucial to take other perspectives into account when referring to spatial oppressiveness and not just focus on how visual components affect oppressiveness. The entire space must instead be taken into account, including the spatial layout, landscape, and architecture. It needs to regard spatial oppressiveness as a product of the interaction between the entire object space and the subject. Additionally, one should consider the effects of the subject's behavior, both psychological and physiological, on oppressiveness. This approach broadens the concept of oppressiveness from visual to spatial, recognizing that oppressiveness is shaped by a dynamic interplay between the environment and the individual's responses.

Many quantitative studies regarding the relationship between spatial elements and oppressiveness have been mentioned in the preceding content. However, there is a gap in research on the relationship between spatial forms and oppressiveness. This study will be grounded in the concept of spatial oppressiveness, with a particular emphasis on the impact of landscape layout on oppressiveness. The study aims to achieve the following objectives: first, this research intends to confirm the applicability of the measurement formula to the spatial oppressiveness within enclosed courtyards based on the proposed notion of spatial oppressiveness and the oppression calculation formula from Morteza et al. [12] Secondly, considering three common landscape layout types (encircling, dispersed, and centralized), the study aims to explore the varying effectiveness of these layout types in alleviating the oppressiveness. Last but not least, through empirical analysis and sample studies, this research aims to uncover the interplay between spatial layout, activities, and oppressiveness. This approach also aims to provide empirical evidence and expand upon existing theories related to oppressiveness.

## 2. Materials and Methods

### 2.1. Study Area

The research site was located within the student apartment courtyard of a university in Beijing, China (Figure 1). The courtyard spans a width of 26.00 m and a depth of 22.54 m, occupying an area of approximately 499.53 m². The eastern, northern, and western sides of the courtyard are surrounded by high-rise buildings with approximate heights of 48 m, while the northern side features a podium building with an approximate height of 15 m. The selection of this research area was based on the following characteristics:

(1) The relatively small courtyard area aligns with the common feature of narrow courtyard spaces found in high-rise environments.
(2) The surrounding high-rise building interfaces in the area meet the necessary criteria for producing visual oppression.
(3) The uniformity of the building interfaces on all three sides facilitates the elimination of the discrepancy of differing façade appearances on oppressiveness (Figure 2).

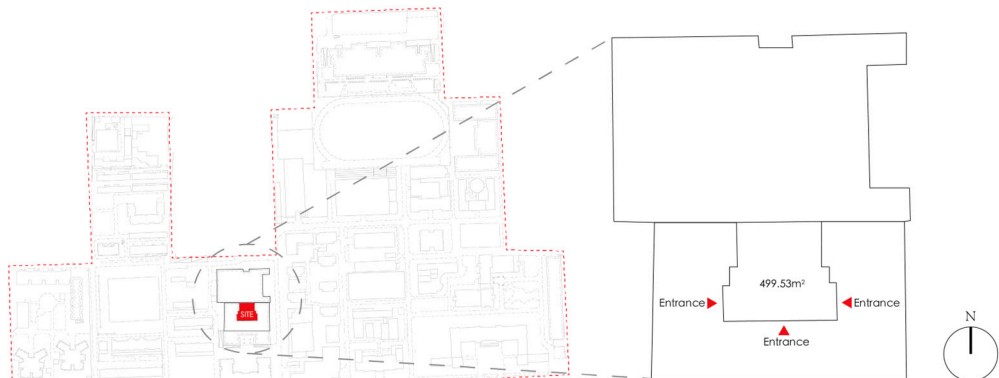

**Figure 1.** Study area.

### 2.2. Sample Design and Scene Simulation

Numerous studies have indicated that factors such as greenery levels [14], architectural façade design [15], and noise [20] can all impact oppressiveness. To minimize potential experimental errors resulting from other factors and to concentrate on analyzing the influence of spatial layouts and human behavior, we maintained a constant ratio of green spaces to usable hard ground at approximately 1.4 in our sample courtyard. Additionally, we ensured that each courtyard comprises a consistent number of five trees, all with uniform heights and canopy widths. As illustrated in Figure 3, and taking into consideration the locations of building entrances and fundamental traffic flow requirements, our final sample design includes the following 4 scenarios:

(1) Control Group Scenario: this simulates the initial, unaltered courtyard with no internal greenery, featuring a hard ground area of 499.53 m$^2$.

(2) Experimental Group Scenario 1: following a green transformation, this courtyard aligns greenery with the building interfaces, concentrating usable hard ground in the central area. The horizontal greenery area measures 318.98 m$^2$, with a hard surface area of 180.55 m$^2$.

(3) Experimental Group Scenario 2: post-green transformation, this courtyard features dispersed greenery arrangements with usable hard ground resembling a grid of road pattern. The horizontal greenery area spans 321.55 m$^2$, with a hard surface area of 177.98 m$^2$.

(4) Experimental Group Scenario 3: after undergoing a green transformation, this courtyard places greenery centrally and on the north side, with usable hard ground forming a U-shaped layout around the central green space. The horizontal greenery area covers 323.33 m$^2$, and the usable hard surface area amounts to 176.20 m$^2$.

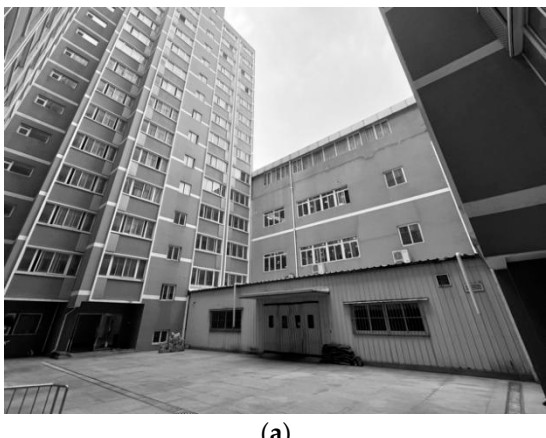 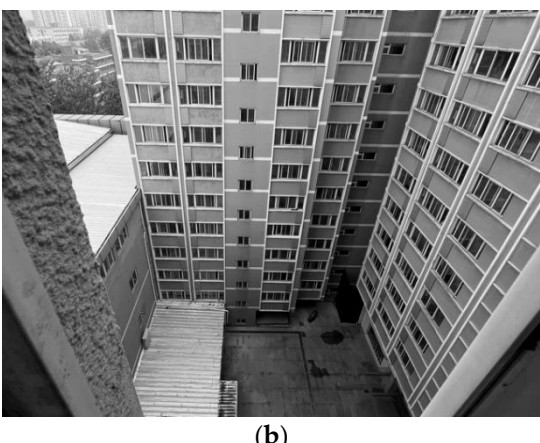

(**a**)         (**b**)

**Figure 2.** Present situation of study area: (**a**) courtyard view and (**b**) high-rise building window view.

*2.3. Research Framework*

In light of the concept of spatial oppressiveness, research should not be confined to quantifying oppressiveness from a single location and a single perspective. It should instead be expanded to measure oppressiveness in the entire space. This entails considering variables such as the willingness of individuals to engage in activities in space, their inclination towards different spatial locations, the overall perception of oppressiveness from freely chosen observation angles, and the impact of spatial atmosphere and landscape imagery on oppressiveness. On top of that, while oppressiveness is a delicate emotional indicator, it frequently calls for a focused and immersed mindset to recognize minute distinctions between various contexts. Therefore, the experimental part of this study consisted of two stages to validate the research hypotheses:

- Experiment One: scene behavior observation experiment. This stage involved simulating pedestrians freely moving within the courtyard. The aim was to investigate how space induces certain behaviors in individuals.

- Experiment Two: oppressiveness measurement experiment. Building upon the results of Experiment One, Experiment Two used the high-frequency travel or stopping points identified in Experiment One as observation points for measuring spatial oppressiveness. Examining oppression from these specific points allows participants to perceive differences more sensitively in the degree of oppression. This approach aimed to obtain more reliable conclusions regarding how individuals perceive differences in oppression across various sample spaces.

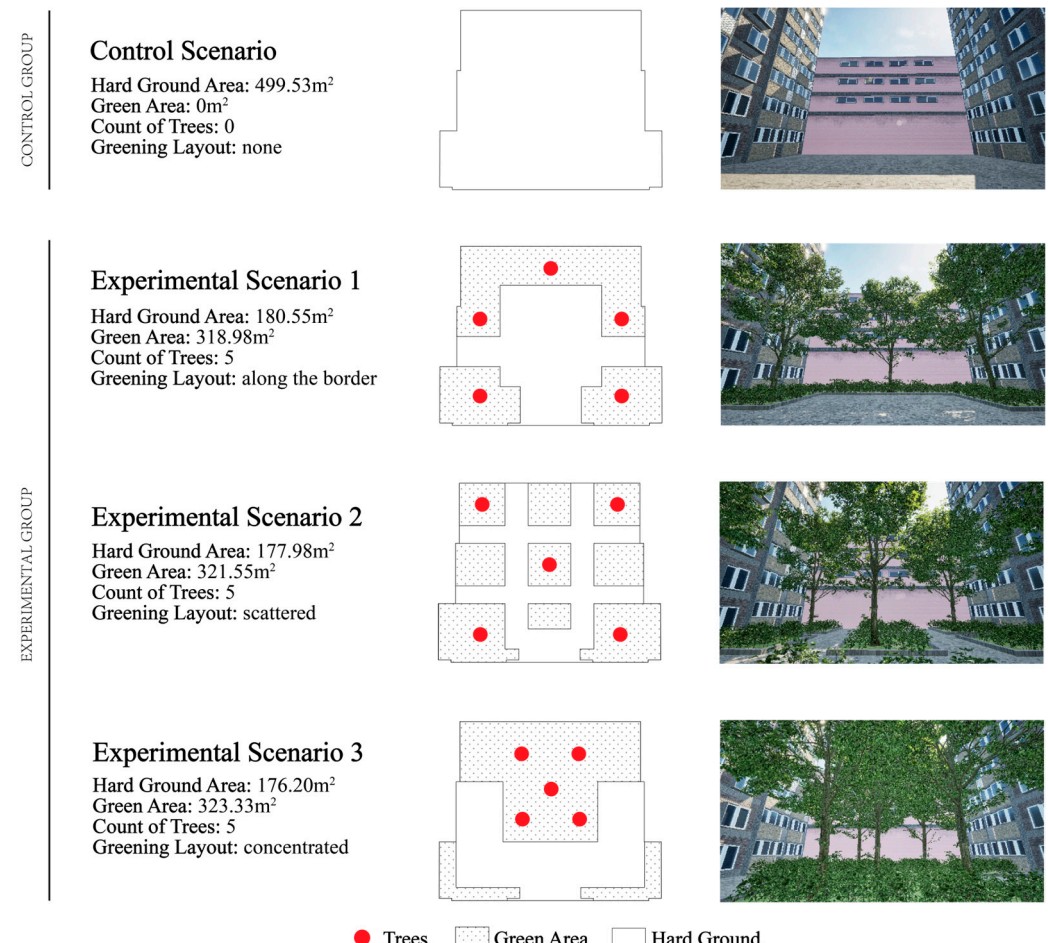

**Figure 3.** Sample design and scene simulation.

Additionally, to compute the theoretical values of spatial oppressiveness for the samples, this study relied on image segmentation on panoramic images together with the oppressiveness calculation formula proposed by Morteza et al. [12]. These values were used for subsequent comparative analyses. By incorporating these methods and considerations, this research aimed to provide a comprehensive understanding of spatial oppressiveness and how it relates to human behavior and perception in complete space.

### 2.3.1. Scene Behavior Observation Experiment

In Experiment One, a behavior observation experiment was carried out as a preliminary investigation into the oppressiveness in the courtyard. The experiment consisted of two main components: the scene-roaming experience and the questionnaire-filling section. In the scene roaming experience section, participants sequentially experienced first-person gaming-style sample scenes built using Unreal Engine 5 (version 5.1) on a high-resolution display screen (3840 × 2160). Participants' movements were confined to the designated activity space within the sample. They had control over the viewpoint's movement using a keyboard and mouse, allowing them to roam freely based on their exploration desires. The program recorded the player's two-dimensional spatial coordinates every 5 s for subsequent analysis. After the experiment was concluded, participants were provided with an electronic questionnaire containing basic demographic information to gather data on the population composition of the experiment.

Experiment One took place from 2 June to 4 June 2023 in the Laboratory of the Department of Architecture at China University of Mining and Technology (Beijing). The participants were 20 campus-based volunteers who were found through online sources. Participants in the trial systematically went through four different sample scenes, each

lasting two minutes. The experiment lasted for roughly 10 min in total. To eliminate potential errors caused by a single experience sequence, the order in which participants experienced the scenes adhered to a Latin square design [21].

2.3.2. Oppressiveness Measurement Experiment

In Experiment Two, a mobile-based panoramic technology combined with electronic SD scale questionnaires was employed. The entire experiment consisted of two phases: the immersive scene experience and the scale-filling phase. Recognizing that the perception of oppressiveness is a highly individualized and immersive process, the scene experience in this experiment was conducted through fixed-point 360° observations by participants. Consequently, the experimental environment was constructed with panoramic technology. In terms of the selection of observation points, this experiment made use of the frequency stopping points identified in Experiment One as observation points. These points represented the locations in the sample courtyard where participants were most likely to linger. Each sample scene had three designated stopping observation points for participants to choose from, allowing them to select their preferred points for the experience. This approach aimed to capture participants' perceptions of oppressiveness from the locations where they were most likely to spend time in the sample courtyard, providing valuable insights into how different spatial configurations impact oppressiveness.

In this stage of the experiment, the SD method was employed as a psychological indicator to gauge participants' perceptions of oppressiveness in the landscape. This method enables a direct evaluation of the experiment's psychological indications. Firstly, the "Oppressiveness" metric was directly included as the primary indicator in this experiment to accurately reflect the actual oppressiveness perceived by participants in the scenes. Secondly, as oppressiveness is a form of stress, the attention restoration theory (ART) proposed by Kaplan and Kaplan postulates that attractive landscapes help refocus one's attention, relieving stress [22,23]. As a result, it was essential to analyze and discuss the indicator "Attraction" in this experiment. Furthermore, the causes of oppressiveness arise from individuals feeling that their personal space in the environment has been encroached upon. Similarly, Bell [24] mentioned that human behavior in spatial environments tends to seek a sense of privacy and territoriality. Places that provide a sense of shelter can stimulate people's desire to stay longer. Therefore, "Territoriality" and "Desire to Stay" were also aspects studied and discussed.

The four-factor axes on the SD scale employed in this experiment were "Oppressiveness", "Attraction", "Territoriality", and "Desire to Stay". Four sets of antonymous adjectives were selected to represent these four factor axes (Table 1). To fulfill the requirements of the quantitative analysis, to each representative scale was given a 5-level evaluation, ranging from 1 to 5. This comprehensive approach aimed to assess and analyze not only the oppressiveness but also the attraction, territoriality, and desire to stay, providing a well-rounded understanding of the participants' perceptions and experiences.

**Table 1.** SD scale.

| Number | Items | Adjectives |
| --- | --- | --- |
| 1 | Oppressiveness | Not oppressive—Oppressive |
| 2 | Attraction | Boring—Attractive |
| 3 | Territoriality | Uneasy—Safe |
| 4 | Desire to Stay | Eager to leave—Eager to stay |

Participants had the opportunity to experience the four sample scenes in a randomized order through an online platform that integrates the scene experience and electronic questionnaires. After experiencing each scene, they rated various indicators in the SD scale matrix for each of the four scenes. This approach helps avoid the issue of non-uniform evaluation scales that may arise when a single scene corresponds to a single evaluation process. Additionally, this streamlined experimental procedure made it easier for participants to

review and evaluate the scenes. Finally, Experiment Two (oppressiveness measurement experiment) took place from 6 June to 13 June 2023. It involved 126 volunteers recruited through online channels, including university students and professionals from various fields, such as architecture and unrelated disciplines. This diverse participant pool ensured that a broad range of perspectives and experiences were considered in the study.

2.3.3. Calculation of Oppressiveness Perception Theoretical Values

In the past, the solid angle has often appeared as a significant parameter in oppressiveness perception calculation formulas, used to describe the size scale of objects observed from a specific point. The calculation formula is: $\Omega = \frac{A}{r^2}$ ($\Omega$ is the solid angle, A is the projected area of the object on the spherical surface, and $r^2$ is the radius of the sphere). Previous research often used 4.5 mm 1:2.8 fish-eye lens photographs as the measurement material for calculating solid angles [12]. However, fish-eye lenses can only capture solid angles in a single direction and may not adequately cover all elements in a three-dimensional environment. In the measurement of spatial oppressiveness perception, it is essential to consider the visual information from the entire space. Therefore, this study used the following method to approximate the solid angles of elements in panoramic spaces:. First, the observation points from Experiment Two were selected as the data collection points for panoramic images, totaling 12 data collection points for the four sample scenes (Figure 4a). UE5's Camera 360 v2 plugin was used at each collection point to capture panoramic images of the environment. Next, the captured panoramic images were processed using Photoshop 2023, categorizing them based on building, sky, plants, and ground and setting color masks (Figure 4b,c). The Pillow (Python library) was then used to batch-process the color distribution of the panoramic images, representing the solid angles of different environmental elements based on the color proportions of different masks. Additionally, the spherical projection scheme for the panoramic images in this experiment was the cube map projection (CMP) (Figure 4d). This projection method maps objects in three-dimensional space onto the six faces of a cube without altering the solid angles occupied by objects in space [25,26]. Hence, it accurately represents the solid angles of environmental elements. Finally, for the three sampling points in each of the four sample scenes, the solid angles of exposed buildings ($\Omega_B - \Omega_{TCB}$) were calculated. Based on the oppression perception calculation formula proposed by Morteza et al. [12]: $\psi = \sum (\Omega_B - \Omega_{TCB})\Gamma^3$ (where $\Gamma$ represents the distance from the observation point to the nearest building interface), the theoretical oppression perception at each observation point was computed.

*2.4. Analytical Framework*

This study's data analysis was divided into two sections: the experimental data analysis and the investigation of the theoretical values derived from the computation of spatial oppressiveness perception (Figure 5). In the analysis of the experimental data, firstly, a descriptive analysis was performed on the behavioral data obtained from Experiment One, and cluster analysis was utilized to characterize the dispersion of coordinate sets, reflecting the landscape's inducement effect on behavior. Secondly, the data obtained from the oppressiveness perception measurement experiment in Experiment Two were processed and analyzed. This involved descriptive statistics of SD scale index scores for each sample to reflect the participants' psychological conditions in various sample scenes, as well as the use of dummy variable regression to analyze the correlation between sample types and various indicators.

The calculation of spatial oppressiveness perception theoretical values was based on panoramic image segmentation and involved statistical results. This section comprises visual descriptive statistics of the segmentation results of panoramic images, descriptive statistics of solid angles in sample spaces, and the calculation of theoretical spatial oppressiveness perception values for each sample.

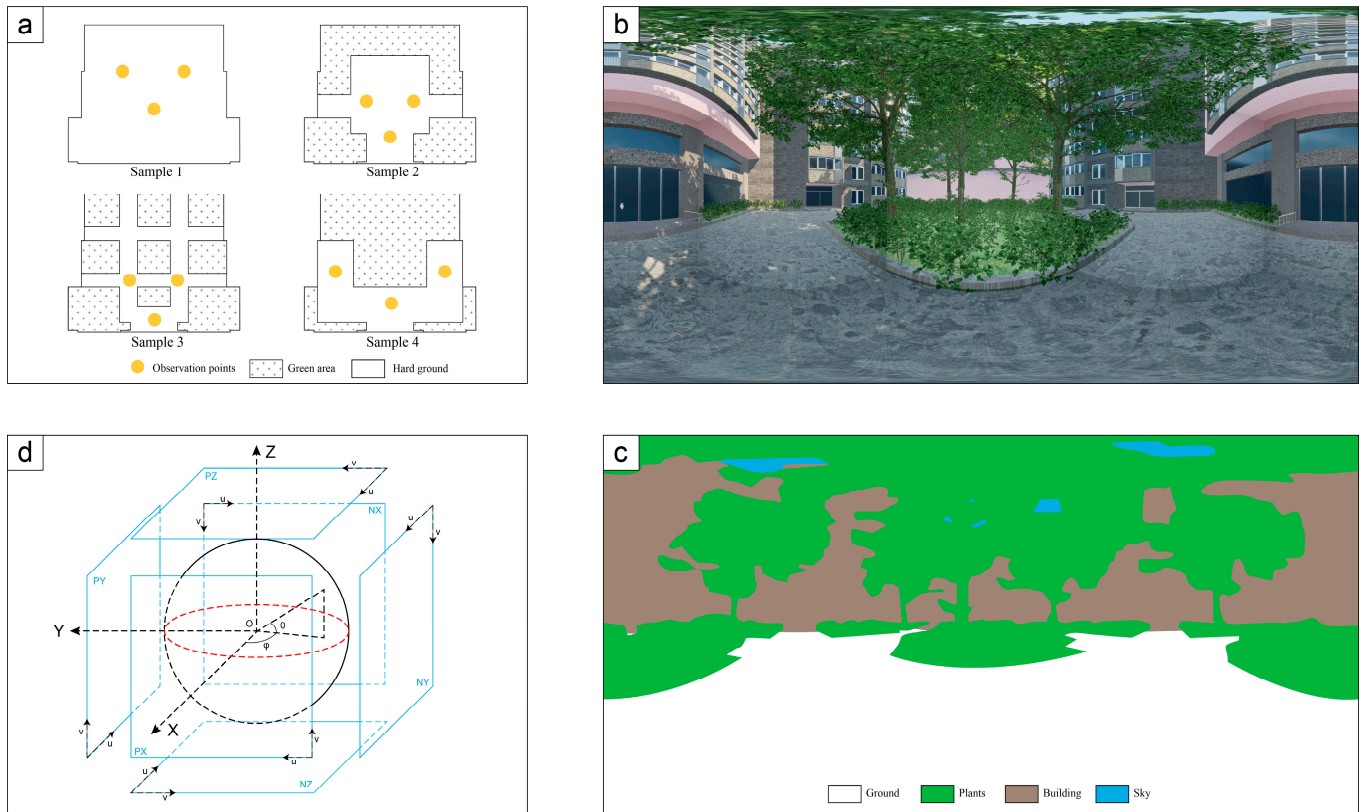

**Figure 4.** Spatial solid angle measurement process: (**a**) sampling point selection; (**b**) panorama image; (**c**) segmentation; and (**d**) stereographic projection.

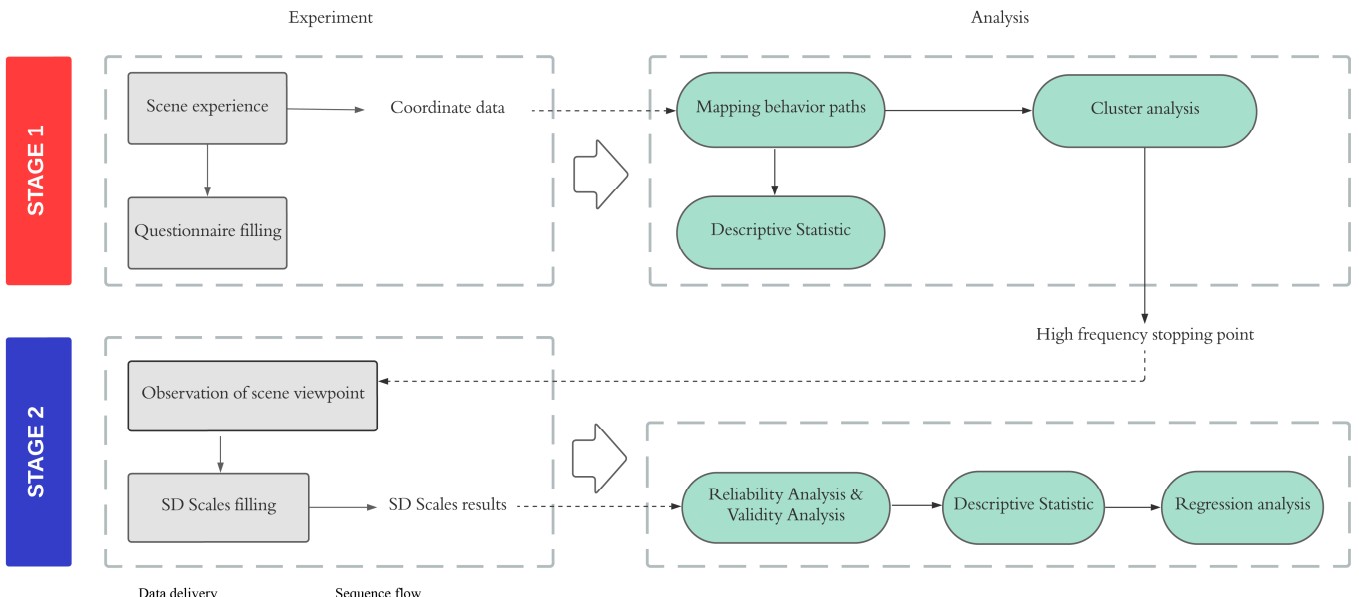

**Figure 5.** Analytical framework.

## 3. Results

### 3.1. Experimental Results

3.1.1. Scene Behavior Observation Experiment Results

In the scene behavior observation experiment, the behaviors of the 20 participants in the scene were recorded in the form of two-dimensional coordinates. Upon the exclusion of human and scene program errors from the coordinate data, the total number of acquired

coordinates for each sample was as follows: sample 1 had 396 coordinates, sample 2 had 352 coordinates, sample 3 had 330 coordinates, and sample 4 had 418 coordinates. Firstly, descriptive statistical analysis was performed on the behavioral coordinate data. The Matplotlib (Python 3.7's library) was used to normalize the coordinate point sets, and the lines connecting the coordinate points were used to represent the behavioral paths of the participants in the sample space (Figure 6). Further, the aggregated characteristics of the behavioral paths were extracted. Based on the scikit-learn (Python 3.7's library), the K-means clustering algorithm was used to perform cluster analysis on the coordinate points in the sample space. This involved dividing the sample coordinates into k clusters and using machine learning to optimize the distances between each coordinate and its cluster center. For the needs of the observation points in experiment two, the k value was set to 3 for cluster analysis. Each sample thus produced three clusters. The analysis results of the behavioral paths are as follows:

- The behavioral paths in sample 1 exhibit significant disorder compared to the other three samples. This demonstrates that participants in the courtyard lack a clear sense of purpose in their behavior or may display a certain degree of disorientation.
- In sample 2, the behavioral paths are mainly concentrated in the central courtyard, maintaining some distance from the surrounding building interfaces and greenery layout. It can also be observed that the behavioral paths tend to converge towards the landscape side (the north side), while there is a clear tendency for the paths to stay away from the non-landscape sides (the south and east–west sides).
- In sample 3, the behavioral paths mainly revolve around the central "sun" shaped road. There is a certain concentration of paths at road intersections and near tree areas. Despite the complex crisscross layout of roads in sample 3, the actual pedestrian paths mostly concentrate on the central road in the courtyard and do not extend much to the surroundings.
- In sample 4, the behavioral paths exhibit a clear concentration in the central area of the courtyard. They also encircle the green areas and show a tendency to extend inward.

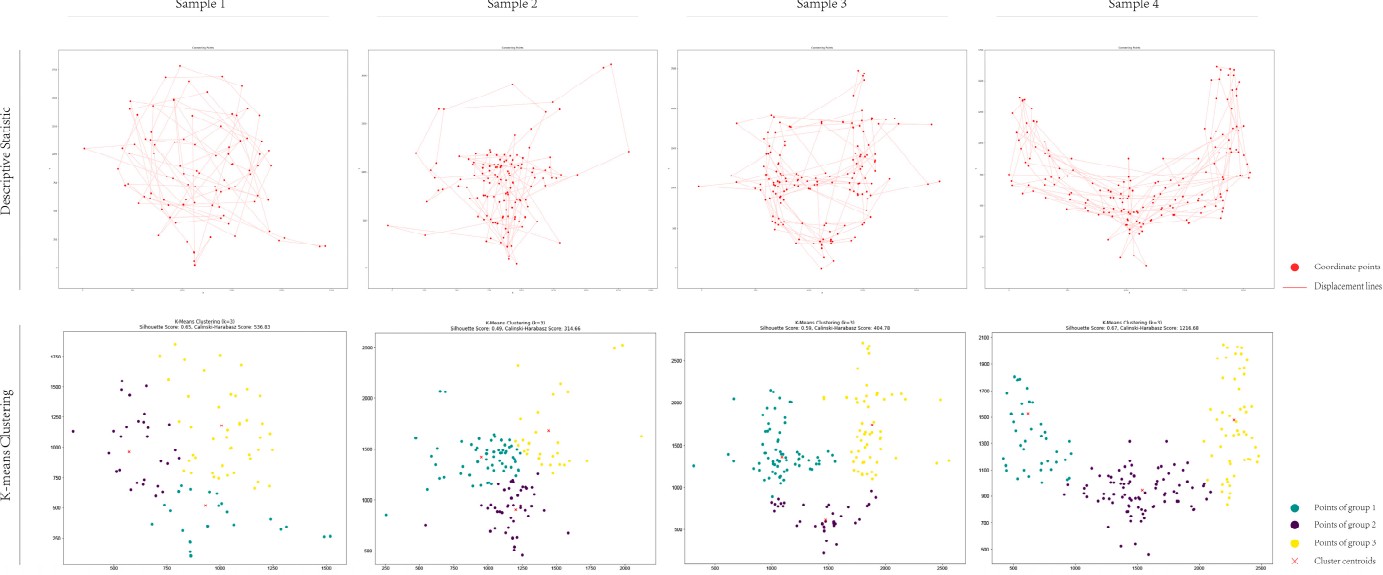

**Figure 6.** Behavior analysis.

The frequency stopping points in the samples are represented by the cluster centroids after clustering (Figure 6, red dots are cluster centroids). Considering the uniformity and directionality required for the observation points in the oppressiveness measurement experiment, adjustments were made on the basis of the clustering results of the behavioral coordinates from various sample scenes in Experiment One. As a result, three observation points were generated for each sample scene in Experiment Two (Figure 7).

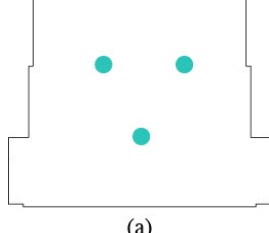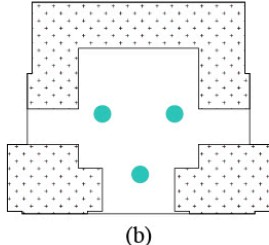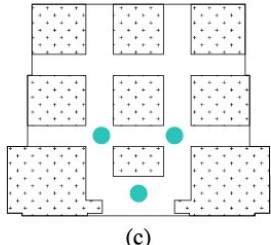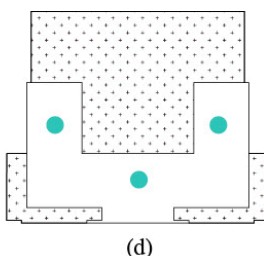

**Figure 7.** Observation points for Experiment Two: (**a**) sample 1; (**b**) sample 2; (**c**) sample 3; and (**d**) sample 4.

3.1.2. Oppressiveness Measurement Experiment Results

The oppressiveness assessment experiment yielded a total of 126 points, and both the recovery rate and effective rate were 100%. A total of 504 original files were imported into SPSS 27 for data analysis. When the SD scale's reliability was examined, the software analysis revealed that Cronbach's α was 0.843, indicating that the questionnaire's reliability was good. Secondly, by one-way analysis of variance (ANOVA) test, we can see that the significance of the four psychological indicators is all $p < 0.001$ (Table 2), which shows that the probability of sample difference caused by sampling error is less than 0.001. At the same time, the F value is also high, which indicates that there are obvious differences between groups. Therefore, further descriptive analysis can be made for the four indicators.

**Table 2.** One-way ANOVA.

| Items | F | Sig. |
|---|---|---|
| Oppressiveness | 82.389 | <0.001 |
| Attraction | 55.921 | <0.001 |
| Territoriality | 74.213 | <0.001 |
| Desire to Stay | 57.897 | <0.001 |

- Part A. Measurement of Spatial Oppressiveness:

"Oppressiveness" will serve as the primary psychological indicator in this experiment and will be utilized to represent the actual spatial oppression experienced by the participants in the sample courtyard. The results (Figure 8a) show that the average "Oppressiveness" score for sample 1 (4.32) is significantly higher than the other three samples, and the data distribution is more concentrated, indicating that participants generally felt significant oppression in the non-green courtyard. sample 4's average "Oppressiveness" score (3.13) is also significantly higher than sample 2 (2.33) and sample 3 (2.44), correspondingly.

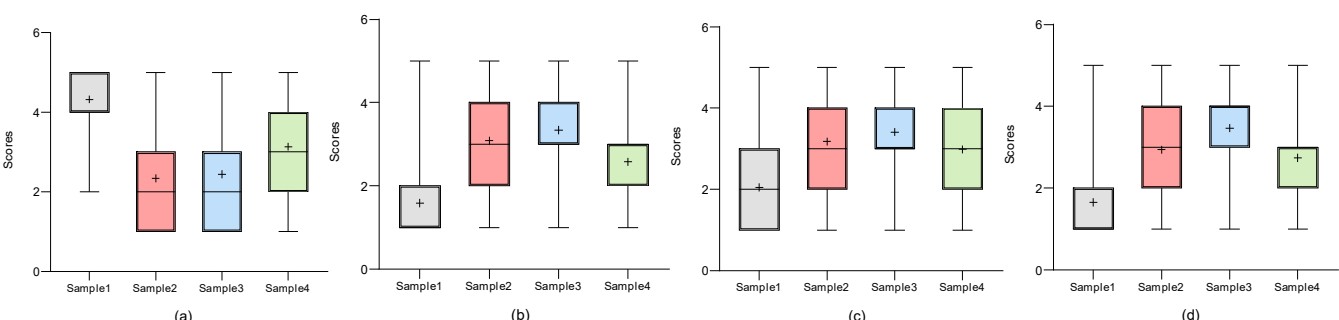

**Figure 8.** Box plot: (**a**) oppressiveness; (**b**) attraction; (**c**) territoriality; and (**d**) desire to stay.

"Attraction", "Territoriality", and "Desire to Stay", as three psychological indicators related to the perception of oppressiveness, will be considered as secondary psychological indicators for statistical analysis (Figure 8b–d). This analysis provides additional insights into the impact of landscape layout on the perception of oppressiveness:

(a) In "Attraction", sample 1 has the lowest score (1.58), while sample 3 has the highest attraction score (3.33), followed by sample 2 (3.08) and sample 4 (2.57). This suggests that non-green high-rise courtyards are less attractive to people compared to green high-rise courtyards, and greenery in the surrounding and dispersed layouts is more attractive than greenery in a centralized layout.

(b) Examining the "Territoriality" index, sample 1 has the lowest score (2.04), while samples 2, 3, and 4 scored 3.17, 3.41, and 3.57, respectively. Analyzing the courtyard layouts, sample 1's lack of landscape elements results in a more exposed space. Comparing this with the behavioral analysis conducted in Experiment One, it can be inferred that natural behaviors make participants feel more sheltered, as seen in samples 3 and 4, which had participants' paths closer to the landscape.

(c) Regarding the "Desire to Stay", the scores for samples 1, 2, 3, and 4 are 1.65, 2.93, 3.46, and 2.74, respectively. This trend aligns with the attraction index results, indicating that a more attractive dispersed landscape layout generates a greater desire to linger compared to surrounding and centralized landscape layouts and no landscape layout.

- Part B. Comparison of Spatial Oppressiveness Relief:

Regarding the correlation research on dummy variables, regression analysis was performed considering the landscape layouts in the sample groups are nominal variables. Sample 1 functioned as the control group, whereas the other sample groups were treated as dummy variables. The dependent variable was the SD scale "Oppressiveness" score, and the data were imported into SPSS 27 for dummy variable regression. In accordance with Table 3's outcomes, compared to the control group with no greenery arrangement, the surrounding greenery layout, dispersed greenery layout, and centralized greenery layout can significantly lessen the participants' oppressiveness. In terms of the magnitude of the alleviation of oppression, the order is as follows: surrounding layout ($\beta = -0.647, p < 0.001$), dispersed layout ($\beta = -0.614, p < 0.001$), and centralized layout ($\beta = -0.388, p < 0.001$). This suggests that the alleviation of oppressiveness by the surrounding and dispersed layouts is similar, while the centralized layout has a much weaker effect in reducing oppressiveness compared to the former two.

**Table 3.** Dummy regression analysis.

| Sample | B | Std. Error | β | t | Sig. |
|---|---|---|---|---|---|
| 1 (Constant) | 4.323 | 0.095 | - | 45.481 | <0.001 |
| 2 | −1.984 | 0.134 | −0.647 | −14.742 | <0.001 |
| 3 | −1.882 | 0.134 | −0.614 | −14.001 | <0.001 |
| 4 | −1.189 | 0.134 | −0.388 | −8.846 | <0.001 |

### 3.1.3. Summary of Experimental Results

The following conclusions specifically for the behaviors of the subjects can be inferred from the analysis of the Experiment One results: firstly, subjects demonstrated a clear tendency to approach landscape elements and move away from tall buildings in unfamiliar environments. Secondly, the behavior of individuals in non-vegetated high-rise courtyards showed signs of disorder and confusion. Lastly, in environments with wider spaces and simpler forms (sample 4), participants exhibited a certain degree of exploratory behavior. However, in narrow and complex environments (sample 3), individuals tended to linger near trees and lacked the desire to explore deeper into the landscape.

Regarding the analysis of the results from Experiment Two and the psychological responses of the subjects, the following conclusions can be drawn: firstly, small-scale courtyards in high-rise areas with vegetation had a significant alleviating effect on oppression compared to those without plants. Furthermore, a surrounding layout and a dispersed layout were more effective in lessening oppression than a centralized layout. Secondly, landscape layouts with higher levels of attractiveness (surrounding and dispersed) were effective in diverting individuals' attention, thereby reducing spatial oppressiveness. These

layouts also influenced individuals' desire to linger in the space. Lastly, participants in the experimental group obtained higher scores for territoriality, indicating that natural landscapes can provide individuals with a sense of shelter, thus alleviating the feeling of oppression within a space.

### 3.2. Results of Oppressiveness Perception Theoretical Values

First, image segmentation was performed on panoramic images collected for each sample (Figure 9). The color scheme used is as follows: blue represents the exposed sky solid angle ($\Omega_S - \Omega_{TCS}$), brown represents the exposed building solid angle ($\Omega_B - \Omega_{TCB}$), and green represents the solid angle occupied by landscape vegetation ($\Omega_T$). From the visual distribution of these color blocks, the following characteristics regarding the distribution of landscape vegetation in the field of view of each sample can be observed. In sample 2, vegetation is situated near the horizon with clusters of plants that are slightly dispersed but exhibit a certain degree of horizontal continuity. Sample 3 displays widespread vegetation distribution throughout the entire field of view with a dispersed clustering pattern. Additionally, the vegetation exerts a significant coverage on the area above the visual horizon. Sample 4, on the other hand, features a relatively concentrated vegetation distribution within the field of view, resulting in weaker coverage on the visual horizon.

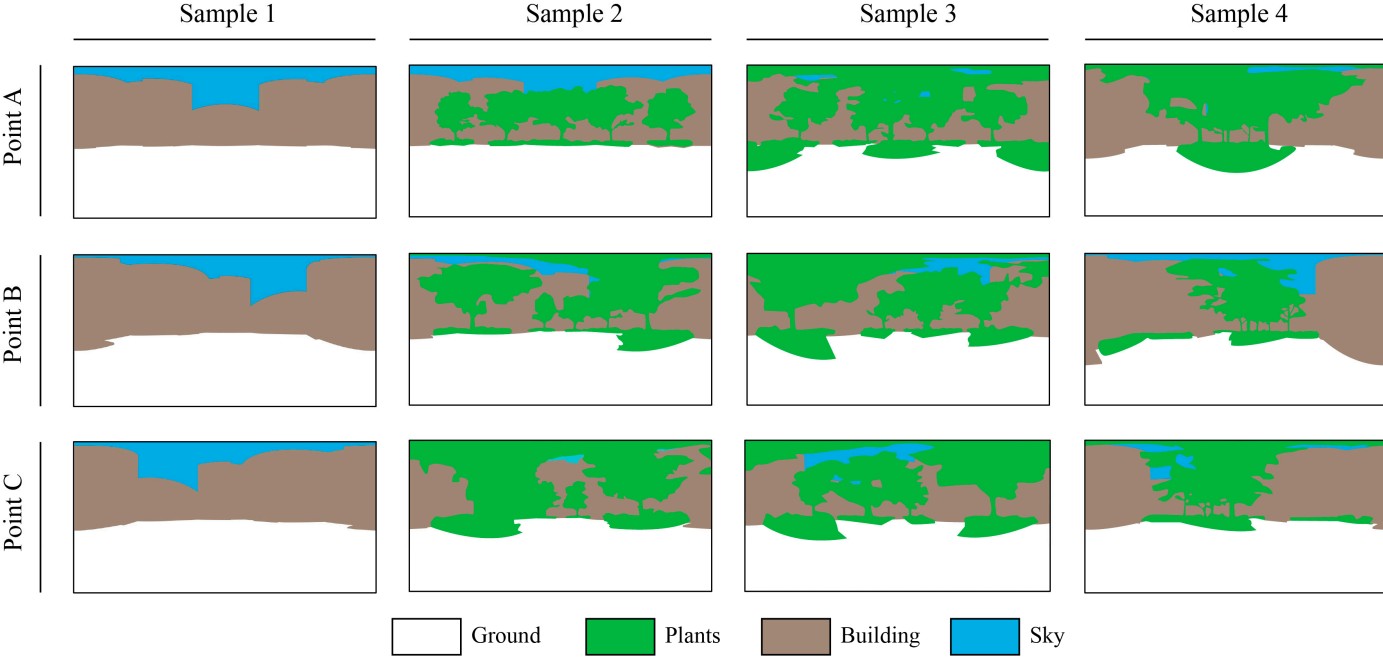

**Figure 9.** Segmentation of panoramic images.

The solid angle ratio was subsequently illustrated by the area proportions of the segmented color blocks in the panoramic images. The average results for the exposed building solid angle ($\Omega_B - \Omega_{TCB}$) at each observation point were as follows: sample 1 was 45.09%, sample 2 was 20.93%, sample 3 was 18.55%, and sample 4 was 32.96%. This indicates that among the experimental groups, sample 3, with its dispersed layout, had the strongest shading effect on buildings, while sample 4, with its centralized layout, had the weakest shading effect on buildings.

Finally, the data on the spatial solid angles at the sampling points and the $\Gamma$ (representing the distance from the observation point to the nearest building surface) were incorporated into the oppressiveness calculation formula proposed by Morteza et al., $\psi = \sum (\Omega_B - \Omega_{TCB})\Gamma^3$, to calculate the theoretical spatial oppressiveness. Since the selection of observation points was determined through K-means clustering, resulting in similar clustering aggregation levels between groups, the average theoretical spatial oppressiveness at the sampling points was used to characterize the overall theoretical spatial

oppressiveness of each sample. The results (Table 4) indicate that the experimental group's theoretical oppressiveness (sample 2, 8.469; sample 3, 5.027; sample 4, 10.272) was lower than that of the control group (15.603). Within the experimental group, the order of theoretical oppressiveness, from highest to lowest, was sample 4, sample 2, and sample 3. This trend in results closely corresponds with the actual oppressiveness reflected in the SD scale results, demonstrating the applicability of the oppressiveness calculation formula $\psi = \sum (\Omega_B - \Omega_{TCB})\Gamma^3$ for calculating spatial oppressiveness in enclosed courtyard spaces.

**Table 4.** Solid angle and theoretical oppressiveness.

| Sample | Point | Solid Angle (%) $(\Omega_B - \Omega_{TCB})$ | Oppressiveness | Mean |
|---|---|---|---|---|
| 1 | A(1) | 47.22 | 26.131 | |
| | B(1) | 44.99 | 10.567 | 15.603 |
| | C(1) | 43.05 | 10.112 | |
| 2 | A(2) | 21.88 | 3.094 | |
| | B(2) | 21.92 | 11.954 | 8.469 |
| | C(2) | 19.00 | 10.361 | |
| 3 | A(3) | 18.77 | 0.621 | |
| | B(3) | 18.20 | 7.138 | 5.027 |
| | C(3) | 18.67 | 7.323 | |
| 4 | A(4) | 28.23 | 2.106 | |
| | B(4) | 38.16 | 13.945 | 10.272 |
| | C(4) | 32.50 | 14.764 | |

## 4. Discussion

Based on the findings from the two experimental phases, it has been demonstrated that various types of landscape layouts can mitigate perceived oppression to variable degrees. The reasons for this difference can be speculated on from the following two aspects: first, the variation in landscape layout leads to differences in visual elements; second, the differences in landscape layout result in variations in behavior intention.

### 4.1. Alleviating Effect of Visual Elements on Oppressiveness

The change in landscape layout significantly affects the distribution of visual elements within a scene. From the results of calculating the spatial solid angle ratio, it is evident that in sample 4, participants are able to observe a substantially higher solid angle for exposed building walls $(\Omega_B - \Omega_{TCB})$, compared to sample 2 and sample 3. This suggests that a centralized landscape layout leads to a higher solid angle ratio of exposed building elements in the scene, making participants more prone to perceiving oppressiveness, which is also consistent with the conclusion of Morteza et al.'s formula for calculating the oppressiveness [12]. Furthermore, based on the element segmentation results in the panoramic images (Figure 9), the tall trees close to the viewer primarily act as shading elements for building facades, playing a crucial role in alleviating the oppressiveness.

Secondly, considering the distribution of color information in the segmented panoramic images, samples 2 and 3 have a dispersed distribution of green vegetation, whereas sample 4 exhibits a more concentrated distribution. Psychologist Fantz [27], through experiments on infant visual perception, demonstrated that humans, from an early age, have a natural inclination toward seeking stimulating experiences and tend to prefer stimuli that are intricate and varied. Consequently, landscape layouts like surrounding and dispersed arrangements, in contrast to centralized ones, offer a more scattered and intricate distribution of landscape stimuli. This, in turn, leads to a heightened preference for the landscape, contributing to the alleviation of a sense of oppressiveness.

Different layouts create variations in the characteristics of courtyard landscapes. Kaplan [22,23] proposed the concept of restorative environments, defining them as environments that enable individuals to recover more effectively from psychological fatigue and

stress. Furthermore, it was indicated that restorative environments generally possess four key characteristics: 'Being away', 'Extent', 'Fascination', and 'Compatibility'. Among the samples in this experiment, sample 2's surrounding layout aligns better with the psychological cognition of a typical courtyard layout. Sample 3's dispersed layout brings people closer to the vegetation landscape, and the spatial pattern is more complex, leading to a higher degree of expansiveness and attractiveness. In contrast, sample 4's centralized layout lacks certain restorative features compared to the former two.

### 4.2. Alleviating Effects of Behavior Intention on Oppressiveness

From the results of Experiment One, it can be observed that different landscape layouts lead to different distributions of movement trajectories in space, with a tendency to converge towards the landscape. This indicates that in a spatial context, individuals tend to behave in a way that brings them closer to nature, away from high-rise interfaces, to avoid the oppressiveness they may cause. Moreover, people's perception of oppression also tends to align with their behavioral inclinations. Taking into consideration the courtyard layouts, the conclusions from Experiment One's behavioral analysis, and the findings from Experiment Two regarding the perception of oppression, it can be inferred that in sample 2, with its surrounding landscape layout, people's activities cluster around the south-facing landscape, away from the high-rise interfaces on the north, east, and west sides. Consequently, the oppressiveness experienced by participants in these two sample scenarios is lower. In sample 4, even though people's activities are consistently concentrated around the central green area on three sides, the central green area occupies the majority of the courtyard's central position. Thus, while participants choose to move towards the landscape, they are unable to move further away from the high-rise building interfaces. Therefore, in the oppressiveness measurement experiment, participants in sample 4 experienced a stronger oppressiveness compared to samples 2 and 3.

Additionally, Appleton's theory of prospect and refuge suggests that humans tend to seek locations where they can "look out without being seen" to find a sense of refuge [28]. Therefore, the inclination of people to gather around attractive landscape features in space can be partly explained by the fact that areas near tall trees or dense vegetation offer a greater sense of refuge. People tend to gravitate towards these psychologically safe spaces to avoid the discomfort associated with exposed areas. This discomfort includes a loss of privacy and the oppressiveness brought about by high-rise buildings. In the sample scenarios of the experimental groups, the arrangement of trees in samples 2 and 3 is more dispersed, whereas in sample 4, the arrangement of trees is more concentrated. This landscape layout provides participants in samples 2 and 3 with more options for psychologically safe spaces, thus alleviating the oppressiveness in these spaces.

### 4.3. Suggestions for Small Courtyard Landscape Design

Based on the conclusions drawn from the above analysis, the following design recommendations are suggested for small-scale courtyard landscape layouts (Figure 10).

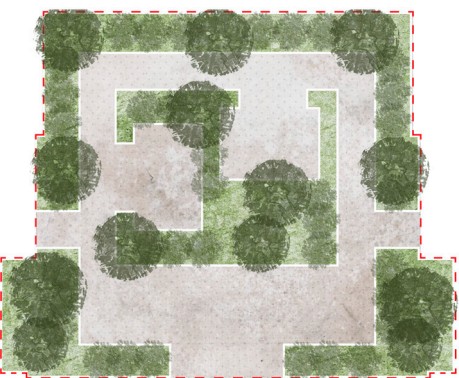

**Figure 10.** Schematic diagram of courtyard design.

Vegetation Placement: strategically position tall trees and shrubs at locations near pedestrian pathways to provide shelter to pedestrians while diminishing the perceived spatial oppressiveness of the courtyard. Moreover, contemplate planting tall trees nearby to the building to provide adequate daylighting to shade any exposed architectural components. Incorporating tall trees as "refuges" in open spaces should be considered to enhance the psychological experience.

Pathways Layout: pathways play a crucial role in directing pedestrian behavior within the courtyard. Clear and well-designed pathways are essential to guiding users effectively. Avoid overly complex layouts that can deter visitors from exploring and using the courtyard. Combining pathways with landscape features can create a more pleasant walking experience and enhance mental well-being.

Activity Space Placement: the activity space refers to the broader hard surface area within a courtyard compared to the roads, providing a place for people to engage in physical activities and relaxation, such as the central hard-surfaced square in sample 2. The placement of activity spaces within the courtyard can significantly impact users' perceptions of visually oppressive elements. Designating activity spaces near the entrance of the courtyard encourages people to engage in activities, promoting positive psychological experiences [29–31]. Ensure that there is some level of landscape separation between activity spaces, providing a balance between privacy and noise reduction.

## 5. Conclusions

This research focuses on small-scale courtyards within high-rise residential areas and aims to analyze the impact and underlying mechanisms of various forms of green layouts on spatial oppressiveness from a comprehensive perspective of three-dimensional space. The study is divided into two halves to address particular questions. In the first phase, we conducted behavioral observation experiments to investigate the inducement of varied landscape layouts on behavior intention. Participants engaged in the exploration of sample scenes aimlessly, yielding valuable data on their behavioral trajectories. In the second phase, we conducted oppressiveness measurement experiments. The experiments were carried out by means of online interactions and questionnaire submissions. For each sample, we obtained data with semantic analysis about spatial oppressiveness and related psychological variables. Furthermore, we presented the concept of spatial oppressiveness and developed a theoretical measurement process based on panoramic image segmentation techniques. Following this methodology, we were able to determine the theoretical values of spatial oppressiveness for the study's sample groups. Through statistical analysis, we have drawn the following conclusions.

This study further corroborates the alleviating impact of trees and landscape greenery on spatial oppressiveness. Even when maintaining the same quantity of greenery, surrounding and dispersed landscape layouts demonstrate a substantially higher ability to mitigate oppressiveness compared to centralized landscape arrangements.

This study employed a method of measuring spatial angles based on three-dimensional projection and panoramic image segmentation. It combined Morteza et al.'s proposed formula for oppressiveness calculation, $\psi = \sum (\Omega_B - \Omega_{TCB})\Gamma^3$, to access theoretical oppressiveness values. The results indicate a relatively small discrepancy between theoretical values and actual measurements, thus demonstrating the applicability of this spatial oppressiveness prediction process in small-scale courtyard spaces.

Landscape impacts spatial oppressiveness chiefly through altering the subject's behavioral tendency and the object's visual elements. Specifically, in terms of visual elements, the shielding degree of the landscape from buildings is the key to relieving oppressiveness, and the higher the shielding degree, the better the effect of relieving oppressiveness will be. Second, a more dispersed layout of landscape stimulus points can also alleviate the sense of space oppressiveness by increasing landscape preference. Lastly, the courtyard landscape layout, which is more in line with the public's psychological cognition and more in line with the public's aesthetic preferences, can also assist in alleviating the oppressiveness to

some extent. In the behavior intention, people will show the behavior intention of being pro-landscape. Hence, the landscape layout with more landscape areas and activity space in the center of the courtyard can bring a lower oppressive effect. Secondly, the space under tall trees can attract pedestrians to seek spiritual shelter, thus resisting the special oppressiveness brought by the courtyard.

Built on the conceptual conclusions of this study, the following recommendations can be provided for the small-scale practice of courtyard design with the purpose of healing: in the aspect of vegetation design, vegetation with human visual height should be arranged near the road as much as possible, and at the same time, tall trees that can provide shelter should be guaranteed in the courtyard. In the aspect of road design, it is necessary to avoid the complexity of the road, and secondly, it is necessary to closely combine the road with the landscape. In the aspect of activity space design, the activity space should be set near the entrance of the courtyard to maximize the spiritual healing effect brought by the activity, and a landscape barrier should be set between the activity space and the building to achieve overall physical and mental health support.

This research has contributed valuable insights into the behaviors of individuals in spatial environments and their perceptions of spatial oppressiveness through a two-phase experimental approach. It nonetheless has several shortcomings that should be noted. The main emphasis of this study was on the landscape layout, specifically the topological relationship between landscape elements and activity spaces (hardscapes) in courtyards. To ensure the clarity of experimental objectives and the singularity of variables, certain specific elements in the courtyards, such as the composition of landscape vegetation and the types of hardscape pavements, were intentionally standardized. Consequently, this study did not consider these specific elements in depth. Furthermore, certain non-artificial errors may be embedded within the experimental results, such as the participants' willingness to engage or individual aesthetic differences. Moreover, given that this study is focused on a specific courtyard case, it lacks a diverse range of scenario studies, such as courtyards enclosed by low-rise buildings or expansive parks. In future research, it would be beneficial to incorporate the study of other specific spatial elements into the realm of spatial oppressiveness and to further expand research on spatial oppressiveness across various types of spaces to broaden and deepen the understanding of factors contributing to its alleviation.

While this research employed specific courtyards within a campus as its case studies, the conclusions drawn from this study hold relevance for the future sustainable development of high-density urban areas. Furthermore, the spatial oppressiveness prediction process proposed in this paper could potentially be applied by urban planners and landscape designers to measure oppressiveness indicators in specific projects, providing valuable guidance for urban development and design assessments.

**Author Contributions:** Conceptualization, Y.C. and L.H.; methodology, L.H.; software, L.H.; validation, Y.C. and L.H.; formal analysis, L.H.; investigation, L.H. and Y.C.; resources, L.H.; data curation, L.H.; writing—original draft preparation, L.H.; writing—review and editing, Y.C.; visualization, L.H.; supervision, Y.C.; project administration, Y.C.; funding acquisition, Y.C. All authors have read and agreed to the published version of the manuscript.

**Funding:** This research received no external funding.

**Institutional Review Board Statement:** Ethical review and approval were waived for this study due to REASON. Firstly, our institution has not set up a relevant institutional review board, so we could not obtain the approval of the institutional review board. Secondly, the research obtained the informed consent of all volunteers. Finally, the article only disclosed the anonymous information of the volunteers.

**Informed Consent Statement:** Informed consent was obtained from all subjects involved in the study.

**Data Availability Statement:** Data are unavailable due to privacy restrictions.

**Conflicts of Interest:** The authors declare no conflict of interest.

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
