# Peer review of "A Study on the Impact of Small-Scale Courtyard Landscape Layouts on Spatial Oppressiveness in Dense High-Rise Environments"

_sustainability, doi:10.3390/su152014826_

Round 1

Reviewer 1 Report

The authors developed a study of great interest and merit. The article shows a good balance between the description of the scope and motivations, methodology, results and discussion. The article is also well organised. Still, this reviewer has some formal concerns, as well as concerns about the content. If the authors are willing, the article can be further improved.

Line 21: consider replacing “stay desire” with “willingness to stay”, “will to stay”, or “desire to stay”.

Lines 37-39: a citation is missing, included in the list of references.

Lines 39-40: references missing for the mentioned studies (to measure and to relieve oppressiveness).

Line 41: Instead of “Takei Masaaki and others” use “Takei et al.” (last/family name only).

Lines 41-44, 44-49, 51-54, 58-63, 63-67, 67-72: from this reviewer's point of view, a better citation practice would be to cite the relevant works of authors next to the mentioned names. Such a technique is even more important when several sentences refer to the same authors’ work.

Line 44: the concept of “building solid angle”, as it is used throughout the article, should be defined in the manuscript.

Line 56: if, when the authors mention Takei, they are citing reference 12, it is a paper with two authors, thus mentioning “Takey and Fukushima” is the correct way of referencing.

Line 63: refer to the author of reference number 13 only by the last/family name.

Line 67: the mention of author Fengyun Cui should be corrected, as only the family name / last name should be mentioned in the text, in the same way it is mentioned in the list of references. Names in all capitals are not acceptable.

Lines 80-82: the authors refer to Heidegger’s work, but they do not cite the author. If another work was consulted referring to Heidegger, then the author should do something like “[Author name] [reference number] referring to the studies of Heidegger, mentions that cognition cannot (…)”. Another option is to cite Heidegger directly.

Lines 83-85: the comment for lines 80-82 also applies to lines 83-85, in this case, about Husserl’s arguments (whose work is not included in the list of references).

Lines 86-88: if the theory is that of Altman’s, why is reference 18 co-authored by another author? The original document that defines Altman’s theory should be mentioned, otherwise, is the co-author also an author of the theory? Is Altman’s theory defined in a chapter of the referenced book? If so, only that chapter should be cited. The authors must be more careful with their citation style.

Lines 99-109: the topic of the paper is only associated with the scope of the journal in terms of social sustainability. Maybe the authors could describe how the aims of their research are related to the scope of the journal. Additionally, what is the underlying objective of studying oppressiveness? Finding ways to alleviate it, although spatial layouts and landscape elements are not the only factors at play? Improving the well-being of people? Disguising oppression? The authors should present a broader motivation briefly.

Line 110 (Materials and Methods section): considering the well-detailed description of materials and methods, the authors could add a subsection referring to the limitations of the study. Although some limitations are referenced in the conclusions, others are still missing. This reviewer advises the authors to refer to the intrinsic oppressiveness of other formal elements, besides the solid angle (e.g. colours, repetitive elements, height of buildings), which are out of the scope of the study (they do not have to be considered, but they can be the focus of future studies). Additionally, the state of mind of the participants may also be a limitation. Participants can be more reluctant or more willing to participate in the experiments and that may affect the results. Moreover, the authors do not refer to the feasibility of growing trees and gardens in highly impervious areas, where the sun hardly shines. Considering the scope of the journal, landscape elements should not be considered mere composition elements. So, the authors should refer to the difficulty of replicability of the virtual pictures in cities as a limitation. The fact that the results can be used in other types of environments (lower buildings, larger parks) should be highlighted (probably in the Discussion and Conclusion sections, but also as a motivation in the Introduction, maybe) if the authors agree with such a statement.

Line 124: include the North direction in the drawing.

Line 144-145: is it a dispersed road pattern or a grid of roads? The idea of a dispersed road pattern is not coherent with the design presented in Figure 3 for Courtyard 3.

Lines 137-151: the Scenarios should have the same name in the text and Figure 3.

Line 151: The authors should indicate that the red dots represent the location of the trees.

Lines 176 and 273: although the work of Morteza et al. has already been cited, the authors must always mention the number of the reference when they cite their work. Include the number of the reference.

Lines 185, 261, 263, 264, 308, 311, 348 and 393: all software, code, libraries and tools should be adequately cited and included in the list of references.

Lines 192-194: this reviewer is assuming that Virtual Reality hardware was used. But was that the case? It should be clear to the reader whether the participants wore VR glasses or were in a room with pictures projected on walls, for instance.

Line 198: a reference is missing for “Latin square design”.

Line 201: the abbreviation SD should be defined when the expression is used first in the text and only once. In this case, it should be done in line 58 and removed elsewhere. After line 58 the authors should only use SD to refer to Semantic Differential.

Lines 225-229: add the reference next to the author’s last/family name. Additionally, the title of the publication does not need to be mentioned.

Line 297: the size of Figure 5 makes it very difficult to read the text in it.

Line 334: it is impossible to read the full data in Figure 6. The axes scale is unreadable, as well as what is supposed to be the cluster analysis data. It is important for that kind of information to be readable, otherwise, the reader only sees a pretty picture without understanding what it means.

Line 342: something went wrong with Figure 7, as this line is blank.

Line 386: readers must be able to read the text in Figure 8. Moreover, the caption refers to a, b, c, d but the charts are not identified accordingly.

Lines 419-420: this conclusion should be better justified, whether by the results or by introducing a reference that refers to this type of service provided by natural elements in built environments.

Line 422: consider rewriting the title of subsection 3.2. Suggestion “Results of Oppressiveness Perception Theoretical Values”.

Line 453: when presenting statistical data, the authors should be careful with the use of the term “significantly” or “significant”. Since the authors are presenting results, they should consider if they are referring to statistical significance or not. If no evidence supports such significance, maybe they would better choose different wording.

Lines 479-480: can the authors please rewrite this sentence? Something seems to be missing. Humans have an innate preference for stimulating environments or for stimulating others? And humans have an innate preference for complex stimuli from an early age?

Line 489: what do the authors mean by “restorative features”? Also, “certain restorative features” is a bit vague.

Line 490: the title of subsection 4.2 seems confusing. Are the authors referring to effects that may alleviate oppressiveness? Are the authors referring to effects that may alleviate behavioural inclinations? Or behavioural inclinations due to oppressiveness? Consider rephrasing the title.

Lines 533-538: how did the results contribute to this suggestion? Although it may be substantiated by the literature, the authors should try to make a more direct connection between their results and the literature (with references), since no actual activity spaces were included in the scenarios. Please improve the justification of this suggestion.

Line 576: what do the authors mean by “and more aesthetic”?

Lines 577 and 582: what do the authors mean by “special oppressiveness”? Is it the specificity of the oppressiveness of the courtyard landscape? If so, “specific” would be more adequate.

Line 580: “better oppressive effect” or higher? Are the authors referring to oppressiveness as a good thing?

Line 16: use “(…) experiments that examine visual, behavioral (…)” instead.

Line 20: use “(…) oppressiveness and supply psychological (…)” instead.

Lines 21-22: use “(…) the visual elements’ solid angles (…)” instead.

Line 44: “angle” is misspelled.

Line 47: a full stop is missing before “Here”.

Line 160: use “On top of that (…)” instead.

Line 217: use “This method enables a direct evaluation (…)” instead.

Lines 227-228: use “(…) can stimulate the people’s desire to (…)” instead.

Line 236: use “(…) of the participants’ perceptions and (…)” instead.

Line 264: use “Pillow” instead of “pillow”.

Line 289: use “(…) reflect the participants’ (…)” instead.

Line 308: use “(…) Matplotlib (…) was used to (…)” instead.

Lines 390-391: use “Sample 1 functioned as (…)” instead.

Line 459: in the third column of the table, the word “angle” is misspelt.

Lines 495-496: use “(…) also tends to align (…)” instead.

Lines 557-558: the first sentence has no subject. Rewrite the beginning.

Author Response

Dear reviewer,

Much appreciated for your suggestions.

In response to your suggestions, we have made the following changes:

  1. Line 21: consider replacing “stay desire” with “willingness to stay”, “will to stay”, or “desire to stay”.
  • Has been revised.
  1. Lines 37-39: a citation is missing, included in the list of references.
  • Line 53
  1. Lines 39-40: references missing for the mentioned studies (to measure and to relieve oppressiveness).
  • The content after this sentence explains the research on measuring oppression; In the next paragraph, the related research on the methods of relieving oppression is explained. Therefore, it is considered not to cite the references in advance here. Line 54
  1. Line 41: Instead of “Takei Masaaki and others” use “Takei et al.” (last/family name only).
  • Has been revised. Line 55 and 58
  1. Lines 41-44, 44-49, 51-54, 58-63, 63-67, 67-72: from this reviewer's point of view, a better citation practice would be to cite the relevant works of authors next to the mentioned names. Such a technique is even more important when several sentences refer to the same authors’ work.
  • Has been revised.
  1. Line 44: the concept of “building solid angle”, as it is used throughout the article, should be defined in the manuscript.
  • The concept of solid angle is further explained in paragraph 2.3.3, so it is considered not to repeat the explanation in the summary part. Line 310-313
  1. Line 56: if, when the authors mention Takei, they are citing reference 12, it is a paper with two authors, thus mentioning “Takey and Fukushima” is the correct way of referencing.
  • Has been revised. Line 55 and 58
  1. Line 63: refer to the author of reference number 13 only by the last/family name.
  • Has been revised. Line 77
  1. Line 67: the mention of author Fengyun Cui should be corrected, as only the family name / last name should be mentioned in the text, in the same way it is mentioned in the list of references. Names in all capitals are not acceptable.
  • Has been revised. Line 81
  1. Lines 80-82: the authors refer to Heidegger’s work, but they do not cite the author. If another work was consulted referring to Heidegger, then the author should do something like “[Author name] [reference number] referring to the studies of Heidegger, mentions that cognition cannot (…)”. Another option is to cite Heidegger directly.
  • Has been revised. Line 97
  1. Lines 83-85: the comment for lines 80-82 also applies to lines 83-85, in this case, about Husserl’s arguments (whose work is not included in the list of references).
  • Has been revised. Line 99
  1. Lines 86-88: if the theory is that of Altman’s, why is reference 18 co-authored by another author? The original document that defines Altman’s theory should be mentioned, otherwise, is the co-author also an author of the theory? Is Altman’s theory defined in a chapter of the referenced book? If so, only that chapter should be cited. The authors must be more careful with their citation style.
  • Co-authors have been supplemented. Line 100
  1. Lines 99-109: the topic of the paper is only associated with the scope of the journal in terms of social sustainability. Maybe the authors could describe how the aims of their research are related to the scope of the journal. Additionally, what is the underlying objective of studying oppressiveness? Finding ways to alleviate it, although spatial layouts and landscape elements are not the only factors at play? Improving the well-being of people? Disguising oppression? The authors should present a broader motivation briefly.
  • The first and last paragraphs of the introduction are supplemented. Line 36-41 and 130-132
  1. Line 110 (Materials and Methods section): considering the well-detailed description of materials and methods, the authors could add a subsection referring to the limitations of the study. Although some limitations are referenced in the conclusions, others are still missing. This reviewer advises the authors to refer to the intrinsic oppressiveness of other formal elements, besides the solid angle (e.g. colours, repetitive elements, height of buildings), which are out of the scope of the study (they do not have to be considered, but they can be the focus of future studies). Additionally, the state of mind of the participants may also be a limitation. Participants can be more reluctant or more willing to participate in the experiments and that may affect the results. Moreover, the authors do not refer to the feasibility of growing trees and gardens in highly impervious areas, where the sun hardly shines. Considering the scope of the journal, landscape elements should not be considered mere composition elements. So, the authors should refer to the difficulty of replicability of the virtual pictures in cities as a limitation. The fact that the results can be used in other types of environments (lower buildings, larger parks) should be highlighted (probably in the Discussion and Conclusion sections, but also as a motivation in the Introduction, maybe) if the authors agree with such a statement.
  • The limitation part at the end of the article is supplemented. Line 723-731
  1. Line 124: include the North direction in the drawing.
  • Has been supplemented. Figure 1.
  1. Line 144-145: is it a dispersed road pattern or a grid of roads? The idea of a dispersed road pattern is not coherent with the design presented in Figure 3 for Courtyard 3.
  • It should be ‘a grid of roads.’ Line 179
  1. Lines 137-151: the Scenarios should have the same name in the text and Figure 3.
  • Has been revised. Figure 3
  1. Line 151: The authors should indicate that the red dots represent the location of the trees.
  • Has been revised. Figure 3
  1. Lines 176 and 273: although the work of Morteza et al. has already been cited, the authors must always mention the number of the reference when they cite their work. Include the number of the reference.
  • Has been supplemented.
  1. Lines 185, 261, 263, 264, 308, 311, 348 and 393: all software, code, libraries and tools should be adequately cited and included in the list of references.
  • The software tools used in this study are UE5 and SPSS, and the programming tool used is Python, all of which are described in this paper and contain the corresponding version numbers. But in the references section, I'm not sure how to refer to these tools.
  1. Lines 192-194: this reviewer is assuming that Virtual Reality hardware was used. But was that the case? It should be clear to the reader whether the participants wore VR glasses or were in a room with pictures projected on walls, for instance.
  • Has been supplemented. Line 223-229
  1. Line 198: a reference is missing for “Latin square design”.
  • Has been supplemented. Line 238
  1. Line 201: the abbreviation SD should be defined when the expression is used first in the text and only once. In this case, it should be done in line 58 and removed elsewhere. After line 58 the authors should only use SD to refer to Semantic Differential.
  • Has been revised.
  1. Lines 225-229: add the reference next to the author’s last/family name. Additionally, the title of the publication does not need to be mentioned.
  • Has been revised. Line 265
  1. Line 297: the size of Figure 5 makes it very difficult to read the text in it.
  • The picture has been enlarged. Figure 5
  1. Line 334: it is impossible to read the full data in Figure 6. The axes scale is unreadable, as well as what is supposed to be the cluster analysis data. It is important for that kind of information to be readable, otherwise, the reader only sees a pretty picture without understanding what it means.
  • The picture has been enlarged, but the article mainly explains and analyzes the overall distribution of coordinates and clustering results (images), without explaining the actual coordinates of coordinate points one by one, so the coordinate axes are not further explained. Figure 6
  1. Line 342: something went wrong with Figure 7, as this line is blank.
  • The picture has been uploaded to figure 7 again.
  1. Line 386: readers must be able to read the text in Figure 8. Moreover, the caption refers to a, b, c, d but the charts are not identified accordingly.
  • The figure has been enlarged and captioned. Figure 8
  1. Lines 419-420: this conclusion should be better justified, whether by the results or by introducing a reference that refers to this type of service provided by natural elements in built environments.
  • This conclusion has been supplemented and modified to some extent. Line 495-498
  1. Line 422: consider rewriting the title of subsection 3.2. Suggestion “Results of Oppressiveness Perception Theoretical Values”.
  • Has been revised.
  1. Line 453: when presenting statistical data, the authors should be careful with the use of the term “significantly” or “significant”. Since the authors are presenting results, they should consider if they are referring to statistical significance or not. If no evidence supports such significance, maybe they would better choose different wording.
  • “significant" has been deleted to eliminate misunderstanding. Line 537
  1. Lines 479-480: can the authors please rewrite this sentence? Something seems to be missing. Humans have an innate preference for stimulating environments or for stimulating others? And humans have an innate preference for complex stimuli from an early age?
  • This sentence has been rewritten. Line 565-571
  1. Line 489: what do the authors mean by “restorative features”? Also, “certain restorative features” is a bit vague.
  • Related concepts have been added. Line 573-576
  1. Line 490: the title of subsection 4.2 seems confusing. Are the authors referring to effects that may alleviate oppressiveness? Are the authors referring to effects that may alleviate behavioural inclinations? Or behavioural inclinations due to oppressiveness? Consider rephrasing the title.
  • This section discusses the inducing effect of courtyard layout on individual behavior, and how this inducing effect can alleviate the oppressive feeling felt by individuals. Behavioral inclinations have been corrected to behavioral intentionality, hoping to reduce misunderstanding. Line 582
  1. Lines 533-538: how did the results contribute to this suggestion? Although it may be substantiated by the literature, the authors should try to make a more direct connection between their results and the literature (with references), since no actual activity spaces were included in the scenarios. Please improve the justification of this suggestion.
  • The definition of activity space in this study is supplemented, and its position in the sample design is explained. Line 643-646
  1. Line 576: what do the authors mean by “and more aesthetic”?
  • "And more aesthetic" has been changed to "and more in line with the public's aesthetic preferences”. Line691-692
  1. Lines 577 and 582: what do the authors mean by “special oppressiveness”? Is it the specificity of the oppressiveness of the courtyard landscape? If so, “specific” would be more adequate.
  • It was an editing error and has been deleted and corrected. Line 692
  1. Line 580: “better oppressive effect” or higher? Are the authors referring to oppressiveness as a good thing?
  • A language error, which has been modified. Line 701

Comments on the Quality of English Language

  • Has been revised.

Thanks much for your review again.

Best wish,

Lianghao huang & Ying Cao

Reviewer 2 Report

The article discuses the relationship between small courtyards nad the spatial oppressiveness. The article explores this relationship from visual, behavioural, and psychological perspectives. As a result of being surrounded by high-rise buildings, small-scale courtyards present a challenge of oppressiveness. I have made some comments that might provide a better version of this manuscript.

1. The abstract is well written. However, I recommend removing vague expressions such as 'holistic perspective.' It needs to be clarified which aspects make it holistic; it dissects visual, behavioural, and psychological perspectives. If so, more descriptions are recommended.

2. The introduction should include references when discussing Japanese scholars. Readers might be interested in refereeing these students' works. Providing other articles that tackle the recent debate about spatial oppression is also essential.

In the introduction, it is crucial to describe the meaning of oppressiveness and spatial oppressiveness.

3. This discussion is well written. However, in the Discussion section, it is essential to link the current results with other studies discussing spatial oppressiveness challenges. These studies discuss its effects on small-scale courtyards' psychological, visual, and behavioural attitudes. I recommend that these research limitations be ignored. I have noticed that the authors mentioned this shortcoming in the conclusion, which is fine with me. It is essential to discuss this research's limitations when using the current research method.

4- The conclusion overlooks the need for future research based on the research limitations. Further research should be conducted to understand the topic better and to uncover any potential drawbacks or benefits of the current study. This would help ensure the research is accurate and has valid results. Additionally, further research should be conducted to identify areas where the recent research can be improved.

The English is okey. 

Author Response

Dear reviewer,

Much appreciated for your suggestions.

In response to your suggestions, we have made the following changes:

  1. The abstract is well written. However, I recommend removing vague expressions such as 'holistic perspective.' It needs to be clarified which aspects make it holistic; it dissects visual, behavioural, and psychological perspectives. If so, more descriptions are recommended.
  • The vague expression has been revised to make it clearer. However, due to the limitation of the number of words in the abstract, it is impossible to further expand the detailed description. Line 13-15
  1. The introduction should include references when discussing Japanese scholars. Readers might be interested in refereeing these students' works. Providing other articles that tackle the recent debate about spatial oppression is also essential. In the introduction, it is crucial to describe the meaning of oppressiveness and spatial oppressiveness.
  • The definition of “oppressiveness” is line 31, and “spatial oppressiveness” is the first viewpoint put forward in this article, therefore there is no previous research to explain it. In order to make the definition of “spatial oppressiveness” clearer, we have supplemented this part. Line 124-125
  1. This discussion is well written. However, in the Discussion section, it is essential to link the current results with other studies discussing spatial oppressiveness challenges. These studies discuss its effects on small-scale courtyards' psychological, visual, and behavioural attitudes. I recommend that these research limitations be ignored. I have noticed that the authors mentioned this shortcoming in the conclusion, which is fine with me. It is essential to discuss this research's limitations when using the current research method.
  • Because the spatial oppressiveness is the concept put forward in this article, the past research has not involved the spatial oppressiveness, so it is difficult to discuss other studies on spatial oppressiveness here. However, we have increased the discussion on the current research results and the past oppressiveness formula. Line 559-560
  1. The conclusion overlooks the need for future research based on the research limitations. Further research should be conducted to understand the topic better and to uncover any potential drawbacks or benefits of the current study. This would help ensure the research is accurate and has valid results. Additionally, further research should be conducted to identify areas where the recent research can be improved.
  • The limitations and the future research directions are further explained. Line 723-731

Besides, we also made other changes to the manuscript:

  • The citation of references has been further revised and improved.
  • The figures have been enlarged to make it more readable.
  • The indicator "desire of staying" has been changed to "desire to stay".
  • The purpose of the study in the introduction has been further supplemented. Line 36-41 and 130-132
  • The method of Experiment 1 is further described. Line 223-229
  • A conclusion has been further revised. Line 495-498
  • The title of subsection 3.2. has been revised to “Results of Oppressiveness Perception Theoretical Values”. Line 499
  • This part has been rewritten. Line 565-571
  • Some grammatical errors have been corrected.

Thanks much for your review again.

Best wish,

Lianghao huang & Ying Cao

Round 2

Reviewer 1 Report

The authors mostly replied to this reviewer's comments satisfactorily.

Nevertheless, and for future reference, many online guides teach how to cite software, code and so on. The authors should have made the effort to search for how to do something. This reviewer agrees that software, code, libraries and tools are correctly mentioned in the text, but adding it to the references list is not that hard and would formally improve the article. Moreover, for instance, Python libraries or code often explicitly mention how it should be cited.

From this reviewer's point of view, Figure 6 should still have been improved.

Nevertheless, the article may be considered for publication.